# Sensory Profile, Consumer Preference and Chemical Composition of Craft Beers from Brazil

**Carmelita da Costa Jardim [1], Daiana de Souza [1], Isabel Cristina Kasper Machado [1,2], Laura Massochin Nunes Pinto [1], Renata Cristina de Souza Ramos [1] and Juliano Garavaglia [1,2,*]**

1    Institute of Technology in Food for Health, University of Vale do Rio dos Sinos, Av. Unisinos, 950, São Leopoldo 93022-000, Brazil; mitajardim@gmail.com (C.d.C.J.); daianasouz@unisinos.br (D.d.S.); ikasper@unisinos.br (I.C.K.M.); lauramnp@gmail.com (L.M.N.P.); rcramos@unisinos.br (R.C.d.S.R.)

2    Department of Nutrition, Federal University of Health Sciences of Porto Alegre, Rua Sarmento Leite, 245, Porto Alegre 90050-170, Brazil

*    Correspondence: julianogar@unisinos.br; Tel.: +55-51-3590-8842; Fax: +55-51-3590-8122

**Abstract:** Craft beers are known for their distinct flavor, brew, and regional distribution. They are made using top-fermenting (ale) yeast, bottom-fermenting (lager) yeast, or through spontaneous fermentation. Craft beers are consumed and produced in Brazil in large quantities. However, they present a high level of polyphenols, which affects consumer preference as they may yield a taste of bitterness to beers. In this study, we analyzed the relationship between polyphenols and bitterness as well as the composition of the main styles of craft beers and consumer preference for them. Six different styles were analyzed according to their polyphenol content, bitterness, chemical composition, sensory profile, and preference. For preference, a panel of 62 untrained assessors was used. For sensory profile, quantitative descriptive analysis was performed using expert assessors ($n = 8$). The most preferred style was classic American pilsner, and the least preferred was standard American lager. The most preferred style showed less bitterness (9.52) and lower polyphenol content (0.61 mg EAG/mL), total solids (6.75 °Brix), and turbidity (7.27 NTU). This beer also exhibited reduced sensory notes of malty, fruity, smoked, hoppy, and phenolic but a higher perception of floral, sweet, and yeast notes; the bitterness attribute had a reduced perception. This study advances the understanding and complexity of the sensory profile of different styles of craft beers from Southern Brazil.

**Keywords:** craft beer; polyphenols; bitterness; preference; sensory attributes

## 1. Introduction

Beer can be defined as a product of cereal fermentation process, and it consists of more than 90% water in addition to carbohydrates, minerals, and alcohol (on average 3.5–10%) [1]. Last year, Brazil produced 13.9 billion liters and consumed 1.25 billion liters of beer, representing 7.0% and 6.6% of the global beer market, respectively [2]. Beers are primarily classified according to the fermentation process [3]. Lagers, the most consumed type of beer, are produced by low fermentation, which is usually carried out between 6 and 15 °C [2]. In contrast, ale type beers are produced by high fermentation, occurring between 16 and 24 °C after which yeast cells rise to the surface of the fermentation media, forming a thick film that is not generally removed completely [2]. Indeed, the transformation of wort into beer essentially represents the yeast-driven conversion of sugars into ethanol, $CO_2$, and many other secondary products that provide specific aromas and flavors [4].

Craft beer can be defined as a distinctively flavored and brewed variety that is distributed regionally. Their popularity has benefited from innovation, creativity, typicality, and authenticity,

which typifies craft beer as an experience that offers pleasure, enjoyment, sense of identity and belonging, self-fulfillment, social recognition, and sustainability [5]. In recent years, there has been a big increase in the Brazilian market for craft beer consumption and production. In addition, craft beer is generally unfiltered, unpasteurized, and without additional nitrogen or carbon dioxide pressure [3]. Unlike commercial beers, craft beers are mainly produced in microbreweries following the basic brewing principles and using specific recipes according to the preference of consumers. At the same time, like commercial beers, they can be brewed using different adjuncts and yeast types [6].

It is well known that many compounds affect the sensory properties of craft beers, such as sugars, organic acids, hop bitter acids, polyphenols, and carbonyl compounds [6]. The fermentation processes through the inoculated yeast (i.e., first fermentation and refermentation in craft beers) are fundamental to the aromatic profile of the final beer produced [4]. Polyphenols are important compounds for beer quality as they can contribute to bitterness, color, body, and astringency and can therefore influence their acceptance [7,8]. Almost 67 different polyphenols have been detected in beers, both from barley and hop [9]. The most abundant phenolic acid is ferulic acid, which is found in different beer styles, especially in pilsner and weissbier [10]. Polyphenols have a key impact on the sensory quality of beers, with a higher number of polyphenols leading to better aroma and flavor of the final product [11].

Beer polyphenols come from barley malt [12] and hop [8], and their content depends on the type of beer and the quantity of hops added during its production. The brewing process and fermentation are also important factors as some chemical changes can occur during these processes [12]. Three polyphenol groups—flavan-3-ol, flavonols, and phenolic acids—are found in beers and contribute to their flavor, aroma, and chemical stability [9]. Some polyphenols act as antioxidants and prevent the oxidative degradation of beers. In addition, they provide potential benefits for human health as they inhibit mutagenic and carcinogenic agents [8].

Consumers choose craft beers because they have a variety of flavors, such as malted barley, chestnut, and honey, which increases the probability of perceiving craft beers to be of a higher quality [13]. Moreover, their consumption has become, in a qualitative approach, experienced-based and emerging from a desire for identity and distinction. The goal toward consumption is not functional but rather symbolic. [5]. Moreover, Brazilian consumers choose craft beers because they have an individual quality value and distinct sensory attributes [14]. Indeed, there has been a worldwide increase in the popularity of craft beers in recent times, particularly traditional ales, lagers, and even styles that do not fit in any of the two main types [3].

In the present study, the relationship between polyphenols and bitterness of the main styles of craft beers brewed in Southern Brazil was analyzed, along with the preference of consumers. In addition, each style of craft beer was characterized according to its chemical composition, polyphenol content, and sensory attributes. As far as we know, few researches have been conducted with the sensorial description and composition of Brazilian artisanal beer styles, evidencing the importance of this work.

## 2. Material and Methods

### 2.1. Craft Beers and Styles

Six different styles of beer were used: standard American lager (SAL), classic American pilsner (CAP), weissbier (WSB), American India pale ale (IPA), Irish red ale (IRA), and robust porter (RPO). Table 1 shows the characteristics and packaging specifications of the different craft beers. These styles were selected so that each specific beer showed different levels of color, bitterness, and ethanol content. All beer sample styles were defined according to sensory characteristics and brewing process as determined by the Beer Judge Certification Program (BJCP) [15]. The beer samples were purchased from the market and were brewed in different localities of Rio Grande do Sul State in Southern Brazil (Table 1).

**Table 1.** Characteristics of each craft beer samples regarding their production and packaging type. Classic American pilsner (CAP), standard American lager (SAL), weissbier (WSB), American India pale ale (IPA), Irish red ale (IRA), and robust porter (RPO).

| Beer Samples | Type | Beer Color | Packing | Packing Volume (mL) | Production City | Purchase Place |
|---|---|---|---|---|---|---|
| CAP | Lager | Yellow | Bottle | 1000 | Porto Alegre | Specialty store |
| SAL | Lager | Yellow | Can | 473 | Caxias do Sul | Supermarket |
| WSB | Lager | Yellow | Bottle | 1000 | Porto Alegre | Specialty store |
| IPA | Ale | Red | Bottle | 500 | Campo Bom | Specialty store |
| IRA | Ale | Red | Bottle | 600 | Porto Alegre | Specialty store |
| RPO | Ale | Brown | Bottle | 600 | Gramado | Specialty store |

### 2.2. Chemical Composition of Craft Beers

For all the beer parameters analyzed, the samples were decarbonated in an ultrasonic bath (Ultra Sonic Cleaner, Unique, São Paulo, Brazil) (30 min and at 80 kHz) until the foam disappeared, indicating that the beer did not contain $CO_2$ [16]. The turbidity was measured in a turbidity meter (TU-2016, Lutron Electronic, Taipei, Taiwan) and expressed in nephelometric turbidity units (NTU). The pH was directly measured using a calibrated pH meter (AZ 86505, AZ Instruments, Taichung City, Taiwan). The total solids were measured by refractometric method using a refractometer (Fisher Scientific, Waltham, MA, USA) and expressed in °Brix.

Dry extract was determined using an aliquot of 25 mL into metallic capsules (weighed before), evaporated in water bath for approximately 30 min, and expressed in g/L. The acidity was measured by titration with a 0.1 M NaOH solution in the presence of phenolphthalein as the indicator until the appearance of pale pink color that persisted for 1 min. The content of reducing sugar was measured using the 3,5-dinitrosalicylic acid method [17]. All procedures were carried out in triplicate, and samples were collected from the same production lot.

### 2.3. Beer Color

The color of craft beers was determined by colorimetric method [18,19]. The color of the beer samples was determined by HunterLAB software and a colorimeter (UltraScan PRO, Hunterlab, Reston, VA, USA) using D65 illuminating standard source, which was calibrated in the ultraviolet region for an accurate measurement of whitening agents. Aliquot of 2 mL of each craft beer was placed in a glass cell with a thickness of 2 mm. The parameters analyzed were luminosity (L*); a* (green to negative value and red to positive value); b* (blue to negative value and yellow to positive value); chroma (C*), which indicates the color purity; and angle measurement (h*), which shows the hue of the samples' color. The C* was calculated by the equation $C* = (a*^2 + b*^2)1/2$); the h* was measured by the equation $h* = tg^{-1}(b*/a*)$. Moreover, the absorbance of beer was measured at a wavelength of 430 nm in a 10-mm cuvette, and the color in European Brewing Convention (EBC) units was obtained by multiplying the absorbance by a given factor [15]. All determinations were carried out in triplicate.

### 2.4. Polyphenols and Antioxidant Analysis

The total phenolic content was determined using the Folin–Ciocalteu method [20]. Briefly, in 500 µL of beer samples or standard solutions, 2.5 mL of 0.2 M Folin–Ciocalteu reagent (Sigma-Aldrich, St Louis, MO, USA) and 2 mL of sodium carbonate (Sigma-Aldrich) solution (75 g/L) were added and mixed. After incubation (2 h), the absorbance was measured at 760 nm. The phenolic content was calculated from the calibration curve of Gallic acid (Sigma-Aldrich) standard solutions and expressed as millimoles of Gallic acid equivalent (GAE) per mL of craft beer. All determinations were carried out in triplicate.

The antioxidant activity was determined by 2,2-diphenyl-1-picrylhydrazyl (DPPH) radical-scavenging activity [21]. A 0.1 mL aliquot of methanolic extract was added to 3.9 mL of a $6 \times 10^{-5}$ mol/L DPPH radical (Sigma-Aldrich), and the absorbance was measured at 515 nm. Quantification was performed

using a calibration curve prepared with Trolox standard (6-hydroxy-2,5,7,8-tetramethylchroman-2-carboxylic acid) (Sigma-Aldrich). The results of DPPH radical-scavenging activity were expressed as µmol of Trolox per mL of beer, and all determinations were carried out in triplicate.

## 2.5. Determination of Bitterness

Craft beer samples were decarbonated, and bitter substances were extracted with iso-octane [16] using 10 mL of sample, 1 mL of hydrochloric acid, and 20 mL iso-octane. After this, the sample was agitated for 5 min at room temperature and then centrifuged for 15 min at 4000 rpm. The iso-octane phase was decanted and drained; the sample tube was covered and left to stand in the dark for at least 30 min before measuring the absorption at 275 nm. Results were expressed as International Bittering Units (IBU), and the average values of three determinations were used.

## 2.6. Sensory Analysis of Craft Beers

Ethical approval for the sensory tests of this investigation was obtained from the University of Vale do Rio dos Sinos Committee (number 12247636), and all participants gave written informed consent to participate in the study. Two different sensory tests—the ranking preference test and quantitative descriptive analysis (QDA)—was performed for each beer style.

For the ranking preference test, a hedonic panel test composed of 62 assessors who were not experienced and aged 20–56 years old was used. The selection criteria were availability and motivation to participate on all days of the experiments and the panelists being regular beer consumers. Initially, these participants answered questions about the habits of beer consumption, such as the frequency; the type, style, and brand consumed; factors that influence consumption (prize, packaging, place of consumption, etc.); sensory characteristics they appreciate the most in craft beers (aroma, flavor, color, taste, foam, etc.); and food pairing with beers. The preference was evaluated by the ranking preference test [22,23]. The test was carried out in individual cabins under white light. In each session, the beer samples were served at refrigeration temperature ranging from 6 °C to 8 °C. About 30 mL of each beer was served in transparent, glass cups without assessors having prior knowledge regarding the brand of the beer being evaluated. The samples were served randomly at the same time, and the assessors were requested to order the least preferred to the most preferred craft beers. The preference tests were carried out in four different sessions with intervals of at least eight hours between sessions to avoid sensory fatigue of the consumers. The results were submitted to Friedman test at a significance level of 5% after which the least significant difference value between the sums of the scores obtained with all analyses was calculated.

For the QDA, the flavor attributes of Southern Brazilian craft beers were analyzed [22,23]. The QDA was carried out by an experienced panel ($n$ = 8) to outline the qualitative aspects of beers. Fifteen attributes, derived from literature, panelists perception, and from the attribute list used by the "beer taster association" [15], were included in the evaluation process. Seven of the attributes were related to flavor (malty, fruity aroma, floral notes, hoppy, phenolic aroma, smoked and yeast odor), two were visual attributes (foam persistency and color), five were gustatory traits (overall intensity, sweet, bitter, alcoholic, residual flavor), and one concerned the texture (level of carbonation). Industrial beers were used in pretesting panel-test sessions to let the assessors familiarize with the products under investigation and the related terminology. These sessions were also used to standardize the panel's attribute definitions according to literature and the panelists' perceptions.

The sensory attributes were assessed using an unstructured 9-point scale anchored at the left end with "absent" and at the right end with "high". The samples were identified with a code of three different random digits, where each panelist received 50 mL of each beer sample, monadically and randomly. In all sensory analysis sessions, the panelists received mineral water and dry unsalted breadsticks for palate cleansing between samples to avoid carry-over effects.

### 2.7. Statistical Analysis

One-way analysis of variance (ANOVA) was performed to detect statistically significant differences among the beers for the sensory attributes and chemical composition. A Tukey honestly significant difference (HSD) post-hoc test was used to identify samples that were significantly different from each other (95% significance). For ranking preference test, the Friedman test and table of Newell and MacFarlane were performed (95% significance). Statistical analysis was done using SPSS Statistics 21 software (SPSS Inc., Chicago, IL, USA). Differences of $p < 0.05$ were considered significant. Principal component analysis (PCA) was carried out on panel QDA data to identify the key attributes that most contributed to the variation in products within the product space. All PCA statistical analyses were performed with the XLSTAT, v2017 package (Addinsoft, New York, NY, USA).

## 3. Results

### 3.1. Craft Beer Composition and Color

All the craft beers tested showed best quality condition parameters according to international quality guidance [15]. Table 2 shows the composition of craft beers. In general, the craft beers had a similar composition in sugars, density, acidity, and pH; more differences were observed in turbidity, total solids, and dry extract.

The porter style (RPO) showed a higher turbidity (230 NTU) than the other tested samples. This beer had a high pH value (4.40), more solids (10% $m/v$), dry extract (7.47 g/L), acidity (2.19 g acetic acid/L), sugars (2.08% $w/v$), and ethanol (7.0% $w/w$). In addition, this characteristic was detected and pointed out by the hedonic panel, which described the beer as turbid and with a dark and intense color, as expected by the analysis of parameters. The SAL exhibited minor turbidity (1.44 NTU), dry extract (3.84 g/L), solids (5.75 °Brix), and acidity (1.49 g acetic acid/L).

**Table 2.** Principal quality parameters of each craft beer. Classic American pilsner (CAP), standard American lager (SAL), weissbier (WSB), American India pale ale (IPA), Irish red ale (IRA), and robust porter (RPO). Different letters in the same column indicate significant differences between groups of beers ($p < 0.05$, ANOVA followed by post-tests).

| Style/Beer | Turbidity (NTU) | pH | Total Solids (°Brix) | Dry Extract (g/L) | Acidity (g Acetic Acid/L) | Density | Sugars (% *w/v*) | Ethanol (% *w/v*) |
|---|---|---|---|---|---|---|---|---|
| CAP | 7.27 [e] | 4.24 [c] | 6.75 [b,c] | 4.20 [d] | 1.84 [c] | 1.0112 [b] | 0.9 [d,e] | 5.1 [c] |
| SAL | 1.44 [f] | 4.12 [c] | 5.75 [c] | 3.84 [e] | 1.49 [d] | 1.0098 [b] | 0.93 [c,d] | 5.0 [c] |
| WSB | 16.78 [d] | 3.88 [d] | 7 [b] | 4.80 [c] | 1.97 [b] | 1.0116 [b] | 0.95 [c] | 5.0 [c] |
| IPA | 37.77 [b] | 4.12 [c] | 7 [b] | 4.21 [d] | 1.97 [b] | 1.0084 [b] | 0.86 [e] | 6.2 [b] |
| IRA | 29.14 [c] | 4.33 [a,b] | 7.75 [b] | 5.36 [b] | 1.52 [d] | 1.0139 [a,b] | 1.13 [b] | 6.2 [b] |
| RPO | 230 [a] | 4.40 [a] | 10 [a] | 7.47 [a] | 2.19 [a] | 1.0222 [a] | 2.08 [a] | 7.0 [a] |

Regarding the color of beers, differences in L*, a*, and b* parameters were found. All samples showed high luminosity, but SAL had higher luminosity than other craft beers analyzed (Table 3). Minor L* value was detected with porter (RPO) style, a very turbid beer (Table 2). The L* value ranged from 14.02 (RPO beer) to 91.65 (SAL beer). The a* value, which represents the color axis green to red, ranged from −0.49 (SAL) to 33.43 (RPO). The positive values indicated a perception of red color due to the toasted barley use in craft beer production. For parameter b*, a tendency of yellow color was noticed and ranged from 24.03 (RPO) to 89.6 (IRA). The decrease in b* value of some samples of craft beers led to a reddish color and with a brown trace, as a function of a* value of color. Chroma value was positive for all craft beer samples and ranged from 32.74 (SAL beer) to 94.19 (IRA). The beer IRA showed a higher chroma when compared to other samples, representing a beer color with more quality, purity, and intensity. The *h* angle oscillated from −1.556 (SAL) to 1.532 (CAP), indicating a more yellow color of beer samples. The *h* is correlated to a* and b* value, and it is important to differentiate the color hue from different beer samples. The CAP beer had a more intense and yellow hue compared to the other samples (Table 3).

**Table 3.** Color parameters of craft beers. L* (luminosity), C* (chroma), h* (hue) and EBC (European Brewery Convention) units. Classic American pilsner (CAP), standard American lager (SAL), weissbier (WSB), American India pale ale (IPA), Irish red ale (IRA), and robust porter (RPO). Different letters in the same column indicate significant differences between groups of beers ($p < 0.05$, ANOVA followed by post-tests).

| Style/Beer | L* | a* | b* | C* | h* | EBC Units |
|---|---|---|---|---|---|---|
| CAP | 87.21 [c] | 1.82 [d] | 47.39 [c] | 47.43 [c] | 1.532 [a] | 13.37 [d] |
| SAL | 91.65 [a] | −0.49 [f] | 32.73 [e] | 32.74 [e] | −1.556 [e] | 7.50 [e] |
| WSB | 89.90 [b] | 1.01 [e] | 40.87 [d] | 40.89 [d] | 1.546 [a] | 9.75 [e] |
| IPA | 77.12 [d] | 12.13 [c] | 71.72 [b] | 72.74 [b] | 1.403 [b] | 16.75 [c] |
| IRA | 62.46 [e] | 29.06 [b] | 89.60 [a] | 94.19 [a] | 1.257 [c] | 44.75 [b] |
| RPO | 14.02 [f] | 33.43 [a] | 24.03 [f] | 41.17 [d] | 0.623 [d] | 157.0 [a] |

The color expressed in EBC units varied from 7.50 (SAL) to 157 (RPO). The beer with higher EBC index (RPO: 157) showed lower luminosity (14.02), and the least intense EBC color had more luminosity (91.65).

*3.2. Bitterness, Antioxidant Activity, and Polyphenols*

The polyphenol content in beer is an important factor to analyze as it can improve the quality and acceptance of craft beers. Table 4 shows the content of polyphenols, antioxidant activity, and bitterness of each craft beer sample. The beers with higher level of polyphenols were RPO (1.62 mg EAG/mL), IRA (0.95 mg EAG/mL), and WSB (1.68 mg/EAL/mL). The SAL craft beer showed lower polyphenols content (0.35 mg EAG/L) compared with other samples. In Table 4, it can be seen that the beers that presented higher content of total polyphenols were also the ones with greater antioxidant activity.

The antioxidant activity was maximal (5.58 μmol Trolox/mL) with the weissbier beer (WSB) using the DPPH method. In general, the antioxidant activity of the tested beers varied from 1.74 μmol Trolox/mL (SAL) to 5.58 μmol Trolox/mL (WSB). The beer bitterness was maximal in IPA beer (46.15 EBU), and the lowest value of bitterness was 9.52 EBU (CAP) (Table 4). The bitterness EBC value varied depending on the content of bitter compounds and polyphenols in the beers. In this case, the craft beer with higher content of total polyphenols did not show a higher bitterness value.

**Table 4.** Total polyphenols content, antioxidant activity (2,2-diphenyl-1-picrylhydrazyl (DPPH) method), and bitterness value of different craft beers. Classic American pilsner (CAP), standard American lager (SAL), weissbier (WSB), American India pale ale (IPA), Irish red ale (IRA), and robust porter (RPO). Different letters in the same column indicate significant differences between groups of beers ($p < 0.05$, ANOVA followed by post-tests).

| Style/Beer | Total Polyphenols (mg EAG/mL) | DPPH (μmol Trolox/mL) | Bitterness (IBU) |
|---|---|---|---|
| CAP | 0.61 [d] | 3.24 [b] | 9.52 [f] |
| SAL | 0.35 [e] | 1.74 [e] | 11.57 [e] |
| WSB | 1.68 [a] | 5.58 [a] | 12.55 [d] |
| IPA | 0.8 [c] | 2.30 [c] | 46.15 [a] |
| IRA | 0.95 [b] | 2.05 [d] | 33.45 [b] |
| RPO | 1.62 [a] | 3.14 [b] | 24.72 [c] |

*3.3. Sensory Analysis of Beers*

For the hedonic test of beers, 62 panelists were recruited to evaluate six different styles. This panel was composed of 57.4% females and 42.6% males. The average consumer age was 32.09 ± 10.6 years old and ranged from 20 to 56 years old. Regarding the frequency of beer consumption, 81.5% of panel reported frequently drinking beer every day and only occasionally consuming it on a weekly basis. They also consumed both commercial and craft beer brands; the most consumed craft beers were local beers, followed by the international brands of craft beers available.

Concerning the factors that influence beer consumption, most assessors chose the beer differential and typical sensory characteristics, the type of serving, the beer label design, and the beer style. The second most important factor was the place of consumption. The least important factor was the type of packaging. The most important sensory characteristics appreciated by the survey participants were the flavor and then the beer fragrance notes. Regarding the preference for styles of beer, the most cited were pilsner, weissbier, and India pale ale.

The calories contained in beer had relevance for only eight participants (12.9%) in the survey, and the clear majority of participants said they usually drink with their friends. When talking about the consumption of beer combined with some type of gastronomic preparation, 24 people (38.7%) reported that they do not care about it, 14 (22.6%) said they do not usually drink with food, and 24 assessors (38.7%) said they try to harmonize the drink with food. Regarding the factors influencing beer consumption, a majority chose the beverage differential, such as how it is served, the label, and the style. The second most important factor was the place where they drink the beer. The least important factor was the packaging. According to sensory characteristics of craft beers, the most prominent was the taste, followed by aroma. For only eight participants (12.9%) in the survey, the calories contained in beer had relevance. On the other hand, the clear majority of participants usually drink with their friends. When talking about the consumption of beer harmonized with some type of gastronomic preparation, 20 people (31.25%) reported that they do not care about it and 14 (22.6%) do not usually drink with the food and only 20 people (31.25%) try to harmonize the drink with the food. Concerning the brewing schools (English, Belgian, German, and American), 63% did not know any of them. Regarding the preference for a particular style of beer, the most cited were pilsner, weissbier, and India pale ale.

Regarding the useful ranking preference test, the least preferred beer was IPA, and the most preferred style was pilsner (CAP). The ranking preference test was considered significant (95% significance) using the Friedman test, and there was a significant difference in the preference when comparing the scores between them. Pilsner craft beer (CAP) was more preferred when compared to lager beer (SAL) and other craft beers. In fact, none of the participants chose CAP beer as the least preferred of all beer samples. Pilsner (CAP), one of the beers with the lowest number of polyphenols (0.61 mg EAG/mL) and bitterness (9.52 IBU), had a higher preference compared to the others. Thus, an increase in polyphenol level and beer bitterness led to a decrease in preference by the panel test. The IPA beer also showed a more intense bitterness (46.15 IBU), which was a factor that contributed to its low preference among beer consumers.

Figure 1 shows the sensory profile of different craft beer styles by QDA. This data indicates the differences in the craft beer styles according to the abovementioned sensory attributes. Aroma attributes, carbonation, hoppy, and foam were some important characteristics evaluated by beer consumers.

The CAP beer showed a high sweet flavor score (3.64) but did not show high scores for other descriptors (Figure 1). The RPO beer exhibited good color (6.23), overall intensity (5.23), foam (5.34), malty (5.08), and smoked (3.48). The hoppiest (4.59) and fruitiest (5.24) craft beer was IPA. This craft beer had around 2.5-fold more hoppy flavor than CAP beer (1.8), which was the most preferred beer. The bitterest craft beer was IPA (7.40) and IRA (7.25). Consumers preferred beers without high polyphenols content, less bitterness (EBU units), and lower bitter and hoppy character.

Differences in the sensory profiles of craft beers were investigated by PCA, and the results are shown in Figure 2. This analysis matrix included all sensory attributes evaluated (Figure 1). Two principal components (PCs) were extracted, and one group of samples was discernible after analysis of PC1 versus PC2 in a biplot of samples and selected variables. In this PCA plot, PC1 explained 37.94% of total variance and PC2 explained another 29.7%. Based on the results of PCA and considering the studied beer samples, CAP, IRA, and WSB were grouped (Figure 2). The beers IPA, RPO, and SAL did not cluster together and remained separated in the plots. The group of beers showed yeast/fermentation and sweet flavor and had a high perception of carbonation. In the upper left

quadrant, the beer IPA was mainly related to the presence of floral flavor. The IPA style showed a more intense perception of floral flavor and hoppy character. The RPO positioned in the upper right quadrant was more related to the presence of more intense color besides alcoholic, malty, fruity, and overall intensity attributes (Figure 2).

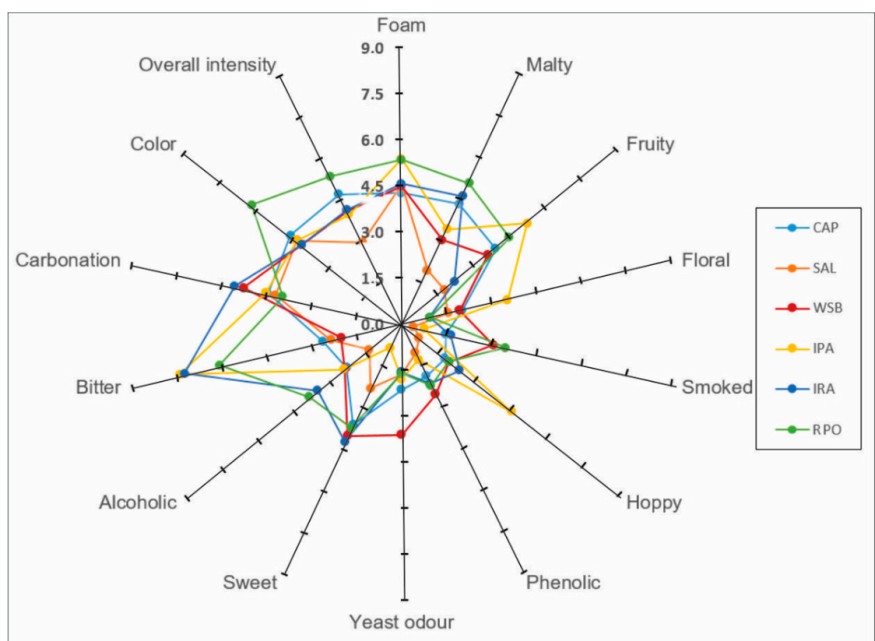

**Figure 1.** Plots of mean intensity scores for sensory profile of six different craft beers evaluated by quantitative descriptive analysis using a 9-point scale. Standard American lager (SAL), classic American pilsner (CAP), weissbier (WSB), American India pale ale (IPA), Irish red ale (IRA), and robust porter (RPO).

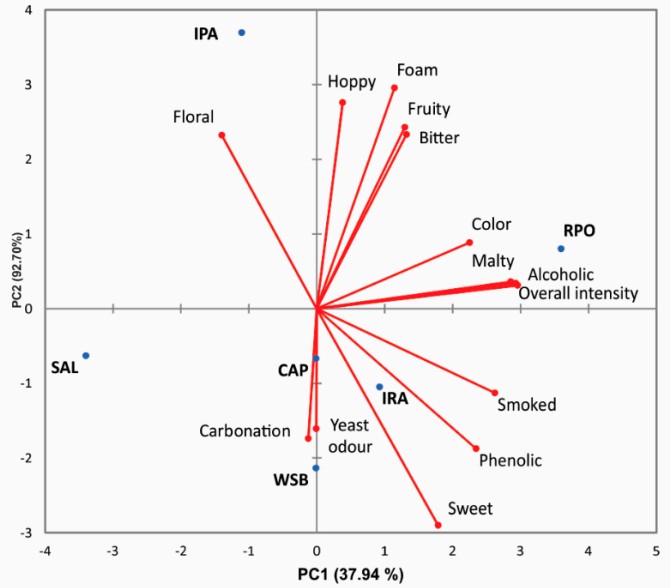

**Figure 2.** Scatter plots of principal component analysis (PCA) scores for specific sensory attributes of Southern Brazilian craft beers analyzed in the present study. (PC1 + PC2 explain 67.64% of total matrix variance). Standard American lager (SAL), Classic American pilsner (CAP), weissbier (WSB), American India pale ale (IPA), Irish red ale (IRA), and robust porter (RPO).

## 4. Discussion

Beer is a very complex mixture, and its chemical composition varies considerably [24], as shown in Table 2. To bring more light into the differences found in craft beer consumption, the objective of this work was to explore the impact of polyphenol content and bitterness of Southern Brazilian craft beers on consumer preference. As craft beers have different flavors, aromas, etc. than the usual well-known commercial brands, their preference is increasing among consumers [13].

These differences in craft beer flavors come from the ingredients used as well as the brewing process [16]. The yeast strains used have a key impact on the final craft beer quality, either in the aroma compound production or in the interaction of some beer components, such as polyphenols [4]. While several yeast strains are commercially accessible, the availability of new starter strains could be an important differentiating factor among craft beers produced in different microbreweries [25]. The main ingredients used in beer production are barley, hops, water, and yeast [26], with each ingredient playing a crucial role in the quality and composition of beers. The porter style beer, for example, is characterized as a substantial, malty dark beer with a complex and flavorful dark malt character [15]. This beer showed high scores in composition parameters compared to other beers tested in this study. Nevertheless, in general, the tested craft beers were similar in analytical factors to the styles described in the BJCP guide [15].

In addition, craft beers have distinctive and pleasant flavor characteristics, and these attributes are easily perceived by consumers [27]. Today, consumer preferences appear to be connected to the discovery of new beer flavors [13], which can increase the consumption of craft beers. Brazilian consumers have followed the same trend as they search for beers with high sensorial quality and differentiated and characteristic flavor and aroma, as verified in this study. The main factor that affects Brazilian beer consumers is the sensory attributes, as pointed out by other studies [13,27,28]. The most preferred craft beer in our study was the CAP style, which mainly shows a fruity and sweet note. Additionally, consumers have a predilection to drink with friends and consider the flavor and fragrances of beer.

It must be noted, however, that this study had some limitations, in particular the few number of craft beer samples that were evaluated for each style of beer. Even so, the sensory attributes and craft beer styles selected in this study for their consumer relevance spanned a wide range of beer characteristics.

Moreover, studying consumer behavior can have great value for the beer industry as it can show how consumers represent the beer category, the associations linked to them, and the proximity across different types of beer [27]. In addition, studies about consumer preferences can assist brewers to understand consumer attitude and translate consumer needs, wants, and expectations into manufacturing design to produce the best, most cost-competitive, and widely accepted product possible in a relatively short period [29].

Beers are rich in polyphenols, which are mostly acquired from barley and hop [8], and they were found in the six styles of beers evaluated in this study. For example, xanthohumol is the most common phenol in hop [30]. Investigating the Brazilian beers, we found that the contents of phenolic compounds, as well as the antioxidant capacity, were like those of beers produced elsewhere in the world [2]. Polyphenols are already extracted in the initial phase of the fermentation process, i.e., during the wort production [9]. This study showed that polyphenols, especially the bitterness associated with it, have an important relationship with the preference of different beers among Brazilian consumers. The most preferred beer showed the lowest bitterness (Table 4, Figure 1) of all styles tested and the second lowest level of polyphenols (0.61 mg EAG/mL). This same relationship with bitterness was verified when analyzing consumer acceptance of craft beers and commercial brands in the Brazilian market [31]. Understanding the sensory characteristic of bitterness in beers and how that relates to their content of polyphenols has a significant value for understanding consumer response as well as optimizing production processes [8].

The malt kilning process determines the color parameter, and it is quite an important process as it can improve the acceptance of beers [26]. The luminosity (L* value) also has a great importance because beers with higher L* value (high luminosity) show a more vivid and intense color [19]. The lager beers show high L* values [19].The darker and more turbid beers show big scores in fruity, floral, and malty flavors, but they still have a low preference among consumers. Beer appearance provides substantial opportunities for product differentiation, and even beers of the same type have the potential to deliver on rather different usage contexts [32].

The most popular beer style in Brazil is the Germany-style pilsner, which is very light and clear [31]. This beer style is very common in the Brazilian market and has a great familiarity to consumers. Familiar beers would be more often cited as appropriate in most of usage contexts, and that familiar and novel products would be associated with different usage contexts [32]. Consumers perceive familiar beers to be more interesting and tasty [32], which can increase their preference, as verified in this study. The preference order obtained from this study came from the sensory proprieties perceived by non-trained assessors because the beer samples were analyzed at the same time and were not assigned different styles.

From the sensory characterization of Brazilian beer styles, it is possible to attest that the evaluated consumers could differentiate and prefer the most aromatic and fruity beers. In addition, this distinct character is a motivation to choose and buy craft beers instead of other beer brands [29]. In this context, the use of *S. cerevisiae* strains isolated from food matrices can represent a valid approach for the selection of starters for brewing to obtain craft beers with more complex flavors [25]. A study with Italian consumers found similar preference to beers brewed from moderately kilned/roasted malts with a milder flavor and less intense mouthfeel perceptions [28]. More complex craft beers with remarkably increased flavors would lead to an improvement of consumer preference [29].

The IPA was the lowest preferred beer and showed a higher level of bitterness attribute perception by the panel test. According to international definitions, the IPA style is a hop-forward, bitter, dryish beer with good drinkability. However, the excessive harshness and heaviness of IPA are typical faults that result from the strong flavor clashes between the hops and other specialty ingredients [15]. Furthermore, IPA beer was differentiated by PCA (Figure 2) from other styles because of its characteristic floral note.

Bitterness is a very important quality parameter in beer production [16]. Nearly four out of ten consumers report that they highly appreciate sweet and fruity samples but dislike primarily bitter, burnt, and roasted notes and hoppy resinousness of beer [28]. The bitter foods are generally disliked due to the instinctive rejection of the bitter taste [33]. Variations in liking and the willingness to consume bitter foods can be triggered by motivational states in humans [33]. In this study, the beer with the lowest bitterness had a higher preference among consumers. This shows bitterness is a key factor that influences beer preference and leads to a decline in consumer preference.

## 5. Conclusions

In this study, we found that the polyphenol content and bitterness determine the preference of craft beers among Southern Brazilian as consumers can perceive their complex sensory attributes. The research also showed the influence of polyphenols in terms of consumer preference as beers with less polyphenol content and bitterness (CAP beer) were more preferred than other craft beer types. Brazilian craft beers with high antioxidant activity, polyphenols, and bitterness were the porter style (RPO), red ale (IRA) and India pale ale (IPA). The craft beers showed complex aromatic notes and flavors, which were described as floral, fruity, yeast, and malty. There were, however, some limitations in this study as it was only exploratory. Therefore, additional work with a larger craft beer sample that is representative of Brazilian craft beers is needed to strengthen our conclusion.

Considering these study findings, it is possible to describe some craft beers and point to the adverse effect of polyphenols and bitterness on consumer preferences. These preliminary results will

be important in stimulating the production of more appreciable craft beers for Southern Brazilian consumers who want to improve their drinking experience and hedonic aspects.

**Author Contributions:** Study conception and design: All; Acquisition of data: C.d.C.J., D.d.S. and L.M.N.P.; Analysis and interpretation of data: All; Drafting of manuscript: C.d.C.J., R.C.d.s.R. and J.G.

**Funding:** This research was funded by Financiadora de Estudos e Projetos (FINEP) of the Brazilian Government, grant number 0110051002. Additional support was provided by Institute of Technology in Food for Health, University of Vale do Rio dos Sinos.

**Acknowledgments:** The authors thank the Financiadora de Estudos e Projetos (FINEP) of the Brazilian Government for financial support.

**Conflicts of Interest:** The authors declare no conflict of interest.

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
