# Peer review of "Sensory Profile, Consumer Preference and Chemical Composition of Craft Beers from Brazil"

_beverages, doi:10.3390/beverages4040106_

Reviewer 1 Report

The manuscript ID: beverages-382308 entitled “Sensory profile, consumer preference and chemical composition of craft beers from Brazil” an article write by Carmelita da Costa Jardim, Daiana de Souza, Isabel Cristina Kasper Machado, Laura Massochin Nunes Pinto, Renata Cristina de Souza Ramos, Juliano Garavaglia is an interesting paper but with some major flaws.

1.       Introduction – needs improvements

·         craft beers types classification

·         rewritten adequately to justify the scope of the work

·         write in a correct English language and use/consulting  recent references

2.       Material and Methods

·         disproportional value among the number of beers samples between Lager (5 samples) and Ale (only one) and between beer color (Table 1)

·         Table 1 – missing the bitterness and ethanol content information (L71) and change POR to RPO (the same in Table 5)

·         The methods was much descriptive – simplify

·         L134 and L173 change de numeration to 2.6 and 2.7

·         L144 – explain who filled out the inquiry- only to the consumer panel?

·         L165- commercial beers – change to industrial beers

·         Statistical analysis – explain why two different statistical programs (SPSS and XLSTAT) were used in the analysis of the data

3.       Results

·         Put 3.1 in 3.4 Sensory analysis of beers

·         L186 - the sample of consumers’ tasters was representative of beers consumers in Brazil? In Brazil women drinks more beer than men??

·         L203-204 – remove the sentence – it was sensorial and not chemical

·         L205 – missing reference

·         L220 – remove sample and beer after RPO and IRA

·         Table 2 – placed before the sentence ….”Regarding the color….

·         Table 2, Table 3 and Table 4 – the beers names (Column) in the same order then table 1.

·         L234-L235 – change the values between parentheses after luminosity

·         L244 – put reference to sustain the sentence

·         2.4 Sensory analysis of beers - to restructure the topic by separating the analysis of the panel consumer from the expert panel trained panel  - to make the text less confusing

·         placed only the Figure 1 or Table 5 - to avoid duplication of the same information

·         L319 – remove “bitter”

·         Figure 2 – remove  1st graphic – scores plots

4.       Discussion

·         L359-367 – remove all the sentence – it is not straight related with this work

·         It is not well explained the interest of the antioxidant activity of craft beers in this work as well as its relation with the sensorial characteristics

5.       References

·         3 references [2, 3 and13] were consulted at 16.01.2017 – explain

Author Response

The response point-by-point was uplaoded in a PDF file.

Reviewer 2 Report

In this article the authors analyze the relation between polyphenols and bitterness of main different styles of craft  beers  brewed  in  Southern  Brazilian,  and  the  preference  of  consumer´s. Phenolic substances is known influence beer quality. Phenolic compounds precipitate with proteins and adsorb yeast cells and stabilization agents and their concentrations decrease during the brewing process. The choice of the raw materials hops and barley malt determines the polyphenol composition in the final beer. Hops have a higher polyphenol content but the greater part of beer polyphenols (70% to 80%) stems from barley malt. Raw materials is changing due to progress in the breeding of barley and hop varieties as well as changing climatic conditions. Moreover yeast strains and their characteristics are important as interact  with polyphenols On the basis of this considerations I agree with the authors that this work have analyzed  few style of beer. I consider the results obtained too restricted, this is a preliminar work that need to be extent to a major number of beers and results are only strictly lied  to Southern Brasilian  regions, reducing international value of the work. On the basis of this limits,  how the results obtained by the authors could be utilised by brewers to produce beer that reflect consumer's expectations? 

Author Response

(The authors gave the same response as above.)

Reviewer 3 Report

This is an interesting paper and I would like to see it published. The craft beer expansion is a worldwide phenomenon and it is interesting to see why and what do consumers really prefer drinking craft beer. 

Here are my comments:

In the Abstract you are mentioning polyphenols as an important factor that influences consumers' preferences. Why would this affect their preference? Health-wise or they contribute to bitterness? 

In the Introduction section you have many English language mistakes and it is very hard to understand the message you are trying to convey. Please refer to an English language expert in order to improve the paper.

"Last year, Brazil produced 13.9 billion liters and consumed 1.25 billion liters, which represented 35 7.0% and 6.6%, respectively, of global beer market." Does this refer to craft or industrial beer or all together? Liters should not be written in whole word, but in abbreviated form (L).

Generally, I do not understand why do you put polyphenols as a selling point for craft beer, please explain. I do not think that an average consumer understand or even thinks of polyphenols while ordering craft beer. I think that they consider craft beer, as you well described in the introduction- "as an experience delivering drink that offers pleasure, enjoyment, sense of identity and belonging, selffulfillment, social recognition, and sustainability". 

I tried to read and understand the results but due to the very poor English language it was very hard to comprehend the sentences and their message. I recognize that the results are interesting not only to scientists but to brewing professional. However, I recommend a proof of English language and style by a professional language service. After that I would really like to read the manuscript and evaluate it. 

Author Response

The response point-by-point was uplaoded in a PDF file.

Round  2

Reviewer 1 Report

Authors responded well to my questions and comments.

However, when I was reading again the new manuscript version (beverages-382308_V2), small details appeared and I suggest changing, if the authors agree (off course)!

Table 1; Table 2; Table 3; Table 4 – in the title of each table change the order of beers designation samples  according to the appearance in the table (Example: 1st CAP designation, after SAL, etc.

Table 3 – complete the title like the others tables (with the mention of style beer)

Material and Methods

· 2.3 and 2.4 – indicate how many replicates were made in these determinations (color; polyphenols and antioxidant analysis)

· you did not changed the point 8 response to reviewer – change the numeration of “Sensory analysis of craft beers” to 2.6 and “Statistical analysis” to 2.7

Results

·  L209-L216 – standardize the sentence - remove the word “beer” after the style beer (example: L212: …14.02(RPO)…

·  L215 – change the sentence to ….”24.03 (RPO) to 89.6 (IRA)

·  L305-L310 – the sentence …”The figure 2….29.7%) put in a paragraph

Discussion

·  L355 – In our work – it was better to use the impersonal form

Author Response

The responses are uploaded in PDF file.

Reviewer 2 Report

The manuscript is improved. Moreover, as  yeast strains and their characteristics are important because they can interact with polyphenols I suggest to the authors to read and cite in the text the chapter  Saccharomyces and non Saccharomyces starter yeasts,  in Brewing technology, INTECH and Novel starter for old precess Marongiu et al, to link the territoriality to craft  beer. 

Author Response

The responses are uploaded in PDF file.

Reviewer 3 Report

Dear authors, 

I see that some improvements have been made regarding English language. However, there are still some mistakes that should be addressed to. I see that the article underwent AJE proofing and jet there are so many cardinal mistakes in some sentences that it is hard to believe this was recommended and proofed by language experts. 

You should check the line numbers since you have big gaps between certain paragraphs (lines 69-72; 191-193, etc.). 

What is missing in this whole story is the amount of hops added to each brew. It is clear that the brew with a generous amount of added hops will have the most polyphenols and relating higher bitterness. This is clearly dependant on the beer style (lagers will have the least hoppy components and ales should contain more hops, especially IPA).

The last sentence before the conclusion section is debatable. Namely, you wrote "...showing that bitterness is a key factor and influences beer preference by consumers." I agree, this is what you discovered in your research, but the sentence does not indicate whether bitterness is something that drives consumers away or attracts them. For example, Czech pilsner beers are always a bit more bitter than German ones, and they still have a wide audience of consumers that are attracted by the bitterness. Also, the season can affect consumer's preference, meaning that hotter climate can drive a consumer into drinking lighter beers (lager or pilsner), and colder regions usually incline toward fuller, dark brews. 

I agree that craft beers are overly bitter. The craft masters put way too much hops and essentially it is hard to taste anything else than hoppy notes. This is why I presume lager and pilsner beers will always stay on top of the consumers preferences. 

Line 241: "The bitterness value varied from content of bitter compounds in beer and not was verified a direct relation of polyphenols content of beers and the bitterness EBC value."  Here you state that polyphenols and bitterness are not in correlation? There are many sentences like these, that are just hard to understand due to the grammatical or style mistakes. 

Line 359: "Polyphenols already occur in the early phase of the brewing process, during wort production [8]."  How do you mean "polyphenols can occur"? You mean they are extracted from raw material into wort? 

Line 371-372: "Nevertheless, we are demonstrated that the beers more dark and turbid showed big scores of fruity, floral and malty flavor, but a small preference". 

Line 378-370: "Consumers perceived familiar beers to be appropriate for most uses, more interesting and tasty [30], which may can to an increase the consumer’s preference, as verified in this study." Poorly written sentence. Please revise. 

General note: there are many of these kind sentences, so please carefully revise the whole paper. Good luck!

Author Response

The point-by-point responses are uploaded in PDF file.
